# Organizational Learning in Healthcare Contexts after COVID-19: A Study of 10 Intensive Care Units in Central and Northern Italy through Framework Analysis

**DOI:** 10.3390/ijerph20176699

**Published:** 2023-09-01

**Authors:** Maddalena Gambirasio, Demetrio Magatti, Valentina Barbetta, Silvia Brena, Giordano Lizzola, Chiara Pandolfini, Francesca Sommariva, Anna Zamperoni, Stefano Finazzi, Silvia Ivaldi

**Affiliations:** 1Department of Human and Social Sciences, University of Bergamo, Piazzale Sant’Agostino 2, 24129 Bergamo, Italy; 2Laboratory of Clinical Data Science, Department of Medical Epidemiology, Mario Negri Institute for Pharmacological Research IRCCS, Villa Camozzi, Via G.B. Camozzi 3, 24020 Bergamo, Italy; demetrio.magatti@marionegri.it (D.M.); valentina.barbetta@marionegri.it (V.B.); stefano.finazzi@marionegri.it (S.F.); 3Independent Researcher, Via Papa Giovanni XXIII 18, Mozzo, 24030 Bergamo, Italy; silviabrena3@gmail.com; 4Independent Researcher, Via Piemonte 5, Alzano Lombardo, 24022 Bergamo, Italy; giordano.lizzola@gmail.com; 5Laboratory of Evolutionary Age Epidemiology, Department of Medical Epidemiology, Mario Negri Institute for Pharmacological Research IRCCS, Via Mario Negri 2, 20156 Milano, Italy; chiara.pandolfini@marionegri.it; 6Independent Researcher, Via Giovanni Paradisi, 2, 20127 Milano, Italy; f.sommariva@comune.cusano-milanino.mi.it; 7Cà Foncello Hospital, Aulss2, Piazzale dell’Ospedale, 1, 31100 Treviso, Italy; anna.zamperoni@aulss2.veneto.it

**Keywords:** organizational learning, intensive care units, COVID-19, framework analysis

## Abstract

The rapid spread of the SARS-CoV-2 virus has forced healthcare organizations to change their organization, introducing new ways of working, relating, communicating, and managing to cope with the growing number of hospitalized patients. Starting from the analysis of the narratives of healthcare workers who served in the intensive care units of 10 hospitals in Central and Northern Italy, this contribution intends to highlight elements present during the pandemic period within the investigated structures, which are considered factors that can influence the birth of organizational learning. Specifically, the data collected through interviews and focus groups were analyzed using the framework analysis method of Ritchie and Spencer. The conducted study made it possible to identify and highlight factors related to aspects of communication, relationships, context, and organization that positively influenced the management of the health emergency, favoring the improvement of the structure. It is believed that the identification of these factors by healthcare organizations can represent a valuable opportunity to rethink themselves, thus becoming a source of learning.

## 1. Introduction

Our era is characterized by constant transformations in the epidemiological scenario that disturb consolidated balances of hospital organization and clinical practice, generating growing uncertainty in carrying out a profession and in the management of health facilities.

Since the declaration on 11 March 2020 by the World Health Organization (WHO), the global health system has had to face the pandemic due to the SARS-CoV-2 virus, a much more difficult challenge compared to the numerous emergencies of recent decades. On 31 December 2019, Chinese health authorities notified an outbreak of pneumonia cases of unknown etiology in Wuhan City (Hubei Province, China). Subsequently, the virus spread to many Asian and European countries, and then to all continents. Italy, in particular, was one of the first countries to be hit hard, starting at the end of January 2020. Within a few weeks, many people contracted the virus, requiring hospitalization in intensive care units (ICU). As a result, hospitals throughout Italy had to quickly reorganize their facilities by increasing the number of intensive care beds and recruiting health personnel to be placed in the wards dedicated to the new, rampant disease [1]. Hospitals had to promptly hire new employees, intercepting young professionals who had recently graduated, recruiting operators who in the past had worked in intensive care and had subsequently migrated to other contexts, or inserting specialists from other disciplines without experience in intensive care. These measures changed the existing balances within organizations, introducing new working, relational, communicative, and management methods.

The current study aims to deepen the theme of organizational learning that healthcare realities have developed in this context. Starting from the analysis of the accounts of several healthcare workers who served in 10 intensive care units in Central and Northern Italy belonging to the GiViTI network (Italian Group for the Evaluation of Interventions in Intensive Care), the factors that influenced the birth of organizational learning during the emergency situation were identified and analyzed.

In the second section, the results of a literature review on organizational learning in healthcare contexts and on the impact that the health emergency had on ICU units and healthcare workers are described. In addition, the theoretical framework has been specified. The third section illustrates the research project, specifying its objectives, the methodology used, and the participants. The framework analysis method [2] was chosen for data analysis since it offers the possibility to explore the perceptions, experiences, and considerations of the participants to identify recurring and transversal themes within the different settings, presenting them graphically in a simple and clear way. The subsequent sections report the presentation and discussion of the results, highlighting the acquired cognitive elements and the prospects for future research.

## 2. Theoretical Framework

### 2.1. The Effects of the Pandemic on Intensive Care: A Look at the Literature

The impact of the spread of COVID-19 on intensive care units is generally analyzed in terms of the criticalities that emerged during the different pandemic waves and of the level of consequences that arose, highlighting difficulties and problems that intensive care workers, hospitalized patients, and family members had to face.

Although intensive care unit workers themselves are more prone to psychopathological symptoms such as anxiety, depression, and post-traumatic stress disorder due to the nature of the intensive care setting [3], the COVID-19 pandemic has had a significant impact on their psychological well-being [4]. In particular, a significant increase in mental disorders, such as severe anxiety, has been found, with a higher percentage in women and nurses than men and doctors [5]; high rates of depression, insomnia, and distress, especially in caregivers in direct contact with patients [6]; burnout and somatic symptoms [7]. Such effects have been compared to symptoms developed by healthcare personnel on the battlefield [8].

In addition, hospitalization and admission to the intensive care unit had a particularly traumatic impact, both physically and psychologically, on patients treated during the pandemic. An increase was found in symptoms related to post-traumatic stress disorder [9], such as insomnia [10,11], anxiety [12,13], depression [14,15], and feelings such as fear and loneliness [16,17]. 

Finally, a significant number of studies have explored the effects of the pandemic on family members of patients admitted to intensive care. These subjects developed symptoms of post-traumatic stress disorder [18] generated by the impossibility of freely accessing hospitals to visit their loved ones and the consequent use of virtual technologies as the only communication tool [19], due to the restrictions put in place by governments to limit the spread of the virus [20]. 

The literature, therefore, seems to give greater emphasis to the harmful effects of the pandemic on individual subjects involved at different levels in intensive care. From the perspective of the psychology of work and organizations, it is interesting to observe how the spread of COVID-19 has impacted intensive care units not only at the individual level, but how this event has modified, altered, and transformed the organizational structure of intensive care itself. By shifting the focus from single variables to a more social, distributed, and discussed interpretation of processes [21], it is possible to highlight how changes in the organization of ICUs have oriented, facilitated, and generated new relational, communicative, management, and professional dynamics among the health professionals involved. In this context, it is possible to underscore the ability of organizations to change, shaping themselves on the needs emerging from the context, thus creating organizational learning.

### 2.2. Organizational Learning as a Social Practice That Is Strictly Related to the Context

Observing phenomena from a socio-constructionist perspective means considering reality as the result of a social process in which individuals belonging to the studied context are involved. In this sense, the organization becomes a socially constructed artifact, created by means of a process of cultural construction and inextricably connected to the places and environment in which it develops. Starting from this, it is possible to understand how organizational learning is characterized by two distinct, but intrinsically connected factors: on the one hand, the knowledge originating from the practices of those engaged in their activities [22], and on the other, a knowledge that is created by the interaction and relationships that individuals build.

Knowledge is understood as a socio-cultural phenomenon, far from an individual and individualistic conception interpreted as the action of a subject who acquires general information from a decontextualized body of knowledge [23]. The act of knowing sees a personal experiential dynamic intertwined with an intersubjective dimension by means of which individuals implement actions in order to enter into a relationship with the context. In this way, a Vygotskian Perspective is taken, that goes beyond the purely cognitive view of learning to embrace a social approach in which organizational knowledge is configured as the set of processes of acquisition and internalization of know-how, knowledge, and practices [24].

The set of behaviors of social actors engaged in interacting with the environment in which they are integrated becomes, therefore, the very foundation of learning, profiling itself as social participation in a practice [25]. 

Therefore, a learning theory that knows how to put the relational, communicative, and professional dynamics of people at the forefront, anchoring them to the context to which they belong, offers the opportunity to understand how these dynamics can create knowledge, becoming not only individual learning, but also organizational learning.

### 2.3. Organizational Learning within Healthcare Settings

Lyman et al. [26] define organizational learning as a process of positive change, in the knowledge, awareness, and collective actions of an organization, starting from experience [27]. Thanks to this process, the organization improves its ability to achieve the desired results through favorable transformations [28]. 

The possibility that this mechanism is implemented by a specific unit belonging to the hospital system is determined by relational and communicative elements and contextual factors. First of all, organizational learning is frequently associated with situations in which the members of a unit share a common intent [29], for example, when they have the same goals to achieve [30] or when they jointly face a crisis [31]. In these contexts, health professionals have the opportunity to share opinions and discuss the strategies to be implemented, presenting different operational and professional viewpoints. A second factor is the motivation that drives group members to improve the performance of the organization when team members are supported, valued, and rewarded by leaders and peers [32]. A third contextual factor that stimulates organizational learning is the presence of psychologically secure relationships characterized by a feeling of trust among unit members [33]. This favors the exchange of ideas and the improvement of working practices, through the carrying out of shared decision-making processes [34], and between different specialists as well [35]. Moreover, if members have had the possibility to work together on several clinical or training occasions, or to work together for a long time, there is a greater likelihood of improved operational efficiency [36] and there are higher rates of organizational improvement [37]. 

The characteristics of an organization, the resources dedicated, human and non-human [38], and the mechanisms that the organization promotes within itself and through comparison with other organizations [35] are also determining factors in the promotion of organizational learning. The hospital’s structure can facilitate and encourage effective interactions between workers, creating a favorable environment for the sharing of ideas, information, and knowledge [39], strengthening motivation and unity among operators [40]. The possibility of comparing clinical practices in briefings and regular meetings [32], for example, leads to a significant increase in knowledge from which the healthcare context benefits. In addition, the hospital structure has an important role in consolidating new knowledge through the standardization of practices [41] and implementing the so-called deliberate learning through demonstrations, meetings, and training sessions. 

Finally, leadership style is significantly associated with the changes that an organization manages to maintain and incorporate in order to improve itself: leaders must form effective teams, creating valid collaborations [42] and encouraging open communication [43] that facilitates change [44]. What has been said so far becomes useful to understand how the emergency caused by the pandemic elicited numerous changes in the dimensions mentioned above, determining new methods, procedures, and practices in the workplace. From this point of view, the present study offers the opportunity to intercept, within the intensive care units studied, the organizational, relational, and communicative elements that may have positively impacted on work environments, generating organizational learning.

## 3. Materials and Methods

### 3.1. Research Objective and Project Phases

This work is part of the ASAP (Antibiotic use during SARS-CoV-2 Pandemic) project, set up by GiViTI and the Institute for Pharmacological Research Mario Negri IRCCS (IRFMN) to describe how clinical practice has changed concerning the use of antibiotics and the management of patients infected in ICUs during the pandemic. GiViTI is a collaborative group founded in 1991 with the aim of promoting and implementing independent research projects in order to evaluate and improve the quality of care within its wards. The first part of the project included a phase of quantitative analysis of data collected from February 2019 to January 2021 with the calculation of specific indicators to compare clinical and microbiological outcomes, such as mortality, prevalence of MDR bacteria, duration and adequacy of antibiotic therapies, and likelihood of developing healthcare-related infections in different types of patients present within the ICUs. All data were collected with MargheritaTre, an electronic medical record developed by GiViTI and IRFMN, which allows description of shared care processes and the integration of clinical, medical, and nursing practice with research projects. The second part is characterized as an activity in support of the first. Through a qualitative analysis, the psychosocial, cultural, professional, and organizational factors that, during the pandemic period, influenced the decision-making processes underlying the management of infected patients were explored. A first phase of semi-structured qualitative interviews with professionals who held positions of responsibility was planned, followed by the creation of focus groups with professionals involved in the management of the emergency in the ICU of the participating healthcare setting. This paper reports the results of qualitative research with a specific focus on organizational learning. The collection of experiences, thoughts, and emotional dimensions of professionals offers the precious opportunity to investigate relational, communicative, and managerial elements present within the ICUs during the health emergency, providing a transversal look at the organizational settings observed. Starting from the accounts of the healthcare workers who served in intensive care during the pandemic period’s various waves, we tried to intercept, mapping their distribution, the relationship, communication, and work characteristics that led to a direction of improvement of hospital settings. These elements were then classified as factors capable of fostering organizational learning.

### 3.2. Participants

This study involved 10 intensive care units from different hospitals in Central and Northern Italy that are part of the GiViTI network. For each center, professionals who worked in intensive care during the different waves were involved: doctors, nurses, healthcare assistants, physiotherapists, infectious disease specialists, microbiologists, and psychologists.

The participating intensive care units belong to the following healthcare institutions: Ospedale di Ospedale Civile San Valentino, Montebelluna;Presidio Ospedaliero San Leopoldo Mandic, Merate (Lecco);Ospedale Maggiore Carlo Alberto Pizzardi, Bologna;Ospedale San Giovanni Bosco, Torino;Ospedale Alessandro Manzoni, Lecco;Ospedale Sant’Anna, Como;Ospedale San Donato di Arezzo;Ospedale Maurizio Bufalini, Cesena;Ospedale Maggiore della Carità, Novara;Ospedale della Misericordia, Grosseto.

The codes used to label the hospitals in the following paragraphs do not follow the order presented above in order to guarantee anonymity with regard to the results obtained. A list containing some demographic information of interview and focus group participants can be found in the Appendix A.

### 3.3. Data Collection

In order to understand the organizational learning experience of the health professionals involved, a qualitative approach was adopted in accordance with the theoretical perspective described in the previous section. This approach offers the possibility to get as close as possible to the different contexts observed through a phenomenological perspective [45], and allows one to grasp the dimensions and dynamics of the change in organizational contexts, intercepting the means and motivations [21].

#### 3.3.1. Interviews

The first phase of the study began in October 2021, with the carrying out of online interviews aimed at gaining an initial, general view of the organizational management, listening to the perspectives of professionals who had leading roles of responsibility during the health emergency. Ten interviews were, therefore, carried out, one for each hospital, identifying a central figure within the intensive care unit who carried out coordination. In these interviews, we mainly investigated the general characteristics and organizational culture distinguishing each work setting, the organization of work in the intensive care units before and during the pandemic, the use of antibiotics in the care of COVID-19 patients, and the major strengths and weaknesses in managing the work during the health emergency.

#### 3.3.2. Focus Group

In a second phase, starting from May 2022, 10 focus groups were organized, one for each hospital involved. Different types of professionals who had worked in the ICUs during the various waves of the pandemic (doctors, nurses, OSS, physiotherapists, infectious disease specialists, microbiologists, and psychologists) participated in the discussion groups. These groups allowed us to delineate different perspectives, permitting a broad overview of the aspects related to the pandemic’s impact on an individual’s job at the emotional, relational, communicative, professional, and organizational levels. 

Due to work needs in the settings involved (shifts and lack of resources), it was not possible to obtain the participation of all the healthcare professionals in each ICU. Moreover, in some settings, certain categories were numerically more represented than others. These biases would seem not to have significantly influenced the results obtained, since the possibility of openly discussing a multitude of professional views led to conversations that were representative of the settings under consideration. The discussion groups were set up to involve two researchers, a discussion facilitator, and an assistant to document the meetings.

The interviews and focus groups were recorded and fully transcribed with participant consent.

### 3.4. Data Analysis

The collected data were analyzed with the framework analysis method developed in the 80s by Jane Ritchie and Liz Spencer and presented in the volume Analyzing Qualitative Data by Bryman and Burgess in 1994 [46]. The method chosen here for data analysis, as outlined by Ward et al. [47], is increasingly used in health research contexts such as midwifery [48], nursing [49], and health psychology [50]. Framework analysis is a form of comparative thematic analysis that uses an ordered structure of inductively or deductively derived themes and emergent themes to conduct cross-sectional analysis using a combination of data abstraction and description [2]. The use of this method permits the identification, description, and interpretation of the key issues present within a single setting and, transversally, between multiple settings, synthesizing and presenting the results through a clean, clear graphic display.

The first analysis of the recordings was purely inductive and allowed the extraction of the key themes in the participants’ accounts. This phase is deeply grounded, basing itself on the observations, experiences, and considerations of the participating subjects who dwell in the contexts studied, in accordance with the socio-constructionist perspective. The second phase of the analysis becomes deductive, starting from the literature on organizational learning in healthcare settings (see Section 2.3). The use of both approaches permits the results emerging from the current study to be inserted within the consolidated literature on the subject, highlighting the specific elements of the study context.

The five phases of the framework analysis are described below, step by step: (1) data familiarization; (2) identifying a thematic framework; (3) indexing of all the study data within the framework; (4) charting to synthesize indexed data; (5) mapping and interpretation of the themes highlighted within the graphs [2].

#### 3.4.1. Data Familiarization

The recordings of interviews and focus groups were initially listened to and transcribed. Transcribing the audio material, albeit long and time consuming, permitted the understanding of the key issues involved in identifying the factors that influence the initiation of organizational learning. Considerations and comments were added to the transcribed material, which was also integrated with the notes taken during the interviews and focus groups. 

The familiarization phase is an iterative process that continues until an initial understanding of the data is achieved [51]. At this stage, we tried, as far as possible, not to take into consideration what the literature said, but to let the data collected in the hospitals speak for itself.

#### 3.4.2. Framework Identification

This second phase had a twofold objective: first, to further understand the data, highlighting similarities and differences; second, to initiate a process of abstraction and conceptualization [2]. In this phase, a framework for data analysis was set up, identifying key issues, concepts, and themes. A priori concepts, derived from the literature knowledge, were combined with what emerged from the data familiarization phase, in order to group, classify, and order the themes [52]. 

Subsequently, the structure that was created was tested on a portion of the data equal to half of the interview and focus group transcripts. This phase, too, involved an iterative process of identifying, deleting, and modifying certain themes.

#### 3.4.3. Indexing

The framework identified in the previous phase was systematically applied to all available data. The topics identified and reported in the analysis structure were connected to the interview and focus group transcripts, starting with the setting up of units of analysis corresponding to the 10 hospitals. It was possible to identify differences and similarities within the single units of analysis and, transversely, between different ICUs. 

#### 3.4.4. Charting

This phase consists of the process of sorting and abstracting data in order to be able to examine them systematically and to create graphic summaries to consolidate or review the decisions taken in the preceding steps with an iterative approach. A table was created, reporting, for each unit, the central themes and their relative subcomponents in order to compare the differences.

#### 3.4.5. Mapping and Interpretation

The final phase of framework analysis consists of mapping the data set to provide an interpretation of the results [52], identifying the factors that may have influenced the materialization of organizational learning during the public health emergency.

## 4. Results

This section lists the results of the five phases of framework analysis described in the previous section. The topics identified in the Data Familiarization phase as facilitating factors for the building of knowledge on the part of hospital organizations are summarized in Table 1.

Table 2 compares the themes obtained from the data set in an inductive and deductive manner, with the relative definitions, and presents the structure that was reconsidered in the Framework Identification phase.

The subsequent Indexing phase did not produce changes to the structure, which was considered adequate.

In the fourth phase, Charting, some labels were modified to ensure greater clarity and some components were eliminated to avoid redundancies. Table 3 shows an example of context No. 1.

Finally, Table 4, built during the last phase, Mapping and Interpretation, presents, in a synthetic and transversal manner, the elements that may have positively influenced the learning development of the healthcare organizations involved. 

## 5. Discussion

Table 4 provides a synthesis showing how the identified factors are distributed in a transversal manner among the hospital settings studied and permits similarities and differences to be identified with respect to the theme studied. The different themes are discussed in a thorough manner in the following paragraphs.

### 5.1. Urge Felt by Healthcare Operators to Do Better

A number of studies have shown that motivation is one of the elements capable of fostering organizational learning [30,32,40]. Similarly, this research highlights how healthcare workers showed a high degree of involvement in managing the public health emergency, putting themselves at the complete disposal of their hospital (“There was a lot of goodwill and the ability that everyone made their resources available to all”. Context 9). A strong determination, leading the participants to stay on the front line to fight the COVD-19 battle, emerged from their accounts. The metaphor of war was frequently used to represent the enormous impact of the pandemic emergency on the healthcare settings and staff of the ICUs (“I tried to feel ready, like a knight waiting for battle, along with other knights, colleagues, nurses, doctors. Ready to fight the pandemic”. Context 3). Despite factors such as the fear of becoming infected and transmitting the disease to loved ones and the fatigue from physically and emotionally strenuous work that continued for many hours every day, healthcare workers felt strongly motivated to continue working and to constantly improve (“Perhaps it was also because we were in the heroic phase, but we were all very proactive: every evening we went home tired, but continued to study”. Context 6). The members of the different ICU groups shared the same goals: when all the staff rows in the same direction, the conditions are created for an organization’s improvement.

This factor was one of the most frequent within the healthcare settings studied, representing a transversally recurring element (9 out of 10). It should be noted that, although this study does not distinguish between the different waves of the pandemic period, this dimension was particularly present during the first pandemic phase.

### 5.2. Safe Psychological Relationships

This study highlighted the beginning of new, and the strengthening of existing personal and professional relationships during the pandemic period (“It has strengthened the relationship between different types of professionals and has done so also from a human point of view: as a group we are much closer, much more united. There was always someone available to help”. Context 1). The sharing of an emotionally and psychologically challenging experience led the staff to rediscover colleagues as a valuable source of support, even feeling like they were a second family (“The colleagues were the ones who supported us. We gave each other strength to carry on. We relied so much on colleagues, the team had become our second family”. Context 4). As pointed out in other studies [33], the presence of safe psychological relationships promotes feelings of trust and respect, and these had a positive impact both emotionally and professionally (“I am happy because I found a second family… a group formed in which you feel really welcomed and, consequently, when you have respect, trust, you work really well”. Context 1). Knowing, and being sure that you can trust, and depend on a colleague leads to a positive atmosphere within the organization (“This experience has united us even more. We shared the same things, the same experiences. I felt fine when I came to work because I didn’t feel alone. A glance was enough to understand each other and, when I couldn’t manage a situation, my colleague would step in for sure”. Context 10). 

This factor also had a frequent distribution (8 out of 10) among the study contexts.

### 5.3. Available Resources

Some research [38] has highlighted how the characteristics of an organization, the dedicated human and non-human resources, impact on the promotion of organizational learning. Similarly, this contribution highlights how the availability or lack of resources affects the possibility that healthcare organizations have to acquire knowledge.

#### 5.3.1. Spaces

The possibility of having adequate space available or of being able to quickly change the structure of the wards was an important element during the pandemic emergency. The relocation and transformation of spaces had to be done in a very short time (“It was a very fast job, in a short time, which made us, let’s say, made us work a lot”. Context 5), based on the current trend of infections.

In some cases (6 out of 10), this factor represented an important element for the management of the emergency, positively affecting planning and the care of patients. Some researches [38,53] have highlighted how the characteristics of an organization, the dedicated human and non-human resources, impact on the promotion of organizational learning.

#### 5.3.2. Materials

The availability of materials such as personal protective equipment (PPE) and technology needed to care for patients infected with the COVID-19 virus became a central factor in patient care. PPEs were essential for protection against contagion, allowing professionals to work safely, both psychologically and literally (“There was never a lack of masks, devices, during any of the waves. We had all the personal protective equipment so we worked confidently”. Context 2). The pandemic emergency also led hospitals to increase the resources available, resulting in a useful arrangement for the future (“We have maintained an increase in the number of opportunities even post COVID, such as beds, devices and medical devices”. Context 5).

This factor, however, was present in few of the settings (4 out of 10). The possibility of finding the necessary material resources, especially during the first wave, was a critical point in the pandemic.

#### 5.3.3. Human Resources

When hospitals were able to increase the number of intensive care workers, the work organization and patient care greatly benefited, creating better conditions for emergency management (“We have had a very large increase in staff. From 22 nurses and 5 healthcare assistants, to 90 nurses and 30 healthcare assistants. This has allowed us to work well and to adequately assist patients”. Context 6). 

This factor was rarely present in the intensive care units observed (2 out of 10).

### 5.4. Experience Working Together

Working closely with one group of people is an opportunity to create more consolidated professional relationships, resulting in a balanced orderliness (“Relationships between new colleagues began to form. We were beginning to get to know each other and therefore began to work better”. Context 6). In their accounts, the participants often use a metaphor relating their work experience to a well-oiled gear that permits the work to be carried out fluidly, improving the work organizations’ efficiency (“At a certain point the gear was created, oiled, in the group we worked extremely well and we therefore all marched to pursue the same goal”. Context 7). In agreement with this, in fact, some studies [36,37] have pointed out that the possibility of working together leads to greater operational efficiency, promoting organizational change.

This factor is infrequent (4 out of 10) due to the extensive recruitment of newly hired health workers or workers from non-intensive care departments. This led to the formation of new teams, without a history of shared work experience.

### 5.5. Multidisciplinary Teams

The possibility of working with different specialists permits an exchange of opinions, perspectives, and professional points of view that enrich the individual. The entry into the ICU of different types of specialists led to the creation of multidisciplinary working groups, favoring patient management methods that were able to take into account multiple aspects (“Networks between hospital specialists have been created. Working on one patient with specialists from different disciplines is certainly valuable”. Context 1). The acknowledgment of the importance of expanding the clinical point of view as an opportunity for the hospital to improve performance and acquisition, and for the specialists working there to gain new, transversal skills and improve individual wealth of knowledge emerged (“It was a period in which we were dealing with pulmonologists, diabetologists, gastroenterologists, cardiologists, hematologists, and they taught us a lot”. Context 6. “Working together with specialists such as psychologists, microbiologists, physiotherapists, on the work organization, but also on patient management, has had a positive impact”. Context 4).

This factor is often present in the participants’ accounts (7 out of 10).

### 5.6. Sharing Points of View to Make Decisions

#### 5.6.1. In General

The lack of knowledge on the course of the disease made it extremely difficult to make clinical decisions with respect to patient management (“Even in this case, sharing knowledge was important, because new drugs, new situations. The exchange was inevitable”. Context 3). For this reason, professionals felt an increasing need to discuss cases (“Before reaching a more targeted treatment decision there is all the teamwork”. Context 8). The possibility of reaching a shared decision was experienced as an element related to overcoming loneliness, lightening the individual specialist’s emotional load linked to decision-making (“The fact that we all met up to talk made the decisions taken every day a bit less burdensome. These meetings also gave us some consolation”. Context 4). Here, as in other studies [34,35], it emerges, therefore, how the exchange of ideas through moments of sharing decisions turns out to be a precious element useful for the birth of organizational learning.

This factor was found in the vast majority of settings (9 out of 10).

#### 5.6.2. Established Moments

As already demonstrated elsewhere [32], constant and scheduled meetings turn out to be invaluable for promoting organizational learning. In this study, many healthcare workers highlighted the frequent presence of briefings held throughout the day, formalizing moments in which each professional could express his opinion (“The morning briefing phase, which we all participated in, in which the cases were analyzed and the process and the clinical path were set out. Everyone participates, everyone points out what the problem is, everyone is largely autonomous, then the problems, the important problems, are focused on”. Context 9).

The possibility of including all the professionals involved in patient care in the decision-making process consented a broader view in terms of the different dimensions of care, positively influencing disease management. Some hospitals understood the effectiveness of a shared decision-making process, officially incorporating the group meetings into their work organization (“The hospital has set up a kind of working group in which there are pharmacists, an intensivist, internists, emergency room doctors, even infectious disease specialists”. Context 8). 

This factor is fairly present in the investigated contexts, distributed in a transversal way (7 out of 10).

### 5.7. Activities to Disseminate Knowledge

The emergency nature of the pandemic period made it necessary to disseminate the new working practices and procedures necessary for containing the virus and for managing patients among intensive care professionals. Some settings developed rapid, efficient ways of spreading knowledge: the creation, use, and dissemination of videos to demonstrate to all colleagues, for example, the new COVID care pathways and the processes for dressing and undressing. These represented useful and very necessary tools (“We had to train everyone well. In the beginning, we made videos to quickly train staff on the dressing procedures”. Context 2). In addition, the entry of professionals with no previous experience in intensive care made it essential to organize activities and training sessions for colleagues by experienced professionals in order to foster rapid learning (“It was my first experience in the critical area, so everything was completely new. The more experienced nurses therefore would therefore make us sit on the floor at 4 a.m. with a tube in our hands, a laryngoscope and would show us how to do it”. Context 6). Indeed, some studies have already highlighted how demonstrations, meetings, and training sessions favor the dissemination of knowledge.

The factor identified is present in less than half of the accounts analyzed (4 out of 10), probably due to the lack of time to devote to training and education.

### 5.8. Standardization of Practices and Procedures

The very high contagiousness of the SARS-CoV-2 virus made it necessary to standardize practices and procedures in order to contain its spread within hospitals. A system of specific paths within the ICU to distinguish *dirty* and *clean* areas was established (“We understood that the main thing was to set up the paths immediately, because once they are there everyone respects them”. Context 2). The differentiation of spaces permitted operators to carry out their work safely, creating a calm atmosphere (“I felt safe because even though I knew there was COVID, we had made protocols, we had drawn up routes. The effort was immense, but we were calm”. Context 3).

The arrangement of the ICUs was reestablished at the beginning of each new wave: with the first wave, it was necessary to improvise, but with the subsequent waves, what had been done previously was repeated, reinstating standardized practices and procedures (“At the beginning we were disorganized. Then, with the second wave, I remembered how to set up the paths, drawing the lines on the floor. During the first wave we improvised, during the second we were all connected”. Context 1). Often, however, the feeling of “always starting from scratch” was perceived. This factor was present in the accounts of the staff from few hospitals (3 out of 10) and highlighted the lack of continuity and long-term management of the emergency on the part of the hospitals’ health management.

### 5.9. Leadership

As previously mentioned, some research [42,43,44] has demonstrated how the leadership style is significantly associated with the changes that an organization manages to maintain and incorporate to improve itself. Similarly, this study highlights how, at different levels, leadership is a central theme for the development of organizational learning.

#### 5.9.1. Hospital Management’s Support

In some settings, the hospital’s health management was able to guide health workers in managing the health emergency, organizing the work, and providing guidelines. This element reassured the healthcare professionals, helping them carry out their work (“From the planning point of view the management’s office was very active. Meetings were frequent so there was always a bit of logic”. Context 5).

In addition, the acknowledgement by the hospital’s health management that the healthcare professionals had a crucial role in making clinical and organizational decisions was fundamental (“The steering committee was managed by our director who relied heavily on us, so the supervisory decisions were not imposed on us from above. There was the intelligence to realize that we were the competent technicians at that point. The question, therefore, was, “Tell me how we can organize ourselves to make your job easier”. Context 10). The involvement of professionals in decision-making, however, was not common (2 out of 10); this factor was, therefore, not very present.

#### 5.9.2. Manager

The possibility of having a manager, an intensive care coordinator with positive leadership skills, helped improve the ward’s management during the healthcare emergency. The participants stressed the importance of the coordinator’s organizational ability (“We were lucky because we had a very strong guide. The nursing coordinator was definitely an unwavering point of reference for the group. She had very strong organizational skills”. Context 10), thanks to which the operators could work in peace. Secondly, the participants also reaffirmed the manager’s relational capacity, specifying that he was able to provide support to the colleagues (“There was a head nurse that I liked very much, because he was able to handle everything, he asked us if we had any problems, if you needed anything. You’d be there getting dressed and he would ask you how you were doing and what you needed”. Context 9).

The data show that strong leadership on the part of the ward manager not only permits the operators to feel acknowledged and to improve their work, but also allows them to become more responsible, favoring improvement in the organization (“There was a leadership in our group that was completely lacking in other settings. The system worked because those responsible coordinated the work well and also had insight as to what to do. I found myself not only appreciated, but also empowered”. Context 5).

The presence of this factor was quite frequent (6 out of 10).

#### 5.9.3. Peer Guidance

Facilitation and support from colleagues was an important factor in carrying out the work. The importance of having an expert colleague guide the group was especially important and that expert gained a leadership role in managing the patients during the different waves of the pandemic, facilitating the work in the ward (“The oldest and most experienced took on the role of shift supervisor to teach those who had no experience in intensive care”. Context 6). At the same time, this role provided emotional support and security to those who felt they had inadequate knowledge and skills (“I never felt abandoned because I always had the opportunity to have a guide behind me that provided the experience I lacked”. Context 5). 

Peer guidance was a fairly present factor in the study’s intensive care units (6 out of 10).

### 5.10. Relations between Hospitals

One of the factors considered important by the participants concerned the relationships that formed, grew, and were maintained during the pandemic emergency. The lack of knowledge on the disease’s progress and the spread of the contagion gave rise to the need for a network between the different health contexts, disseminating information on therapies, antibiotics, and practices that had had positive results in caring for COVID-19 patients (“One of my colleagues was the one who frantically visited all the ICUs, asking to be informed, trying to share all the most meaningful, most important choices”. Context 2). Doctors and nurses participated in webinars and meetings with hospitals that had been hit early by the pandemic, in order to be able to best arrange their set up (“We participated in webinars with the other Lombardy Region ICUs that had first been hit, i.e., Bergamo, Pavia, Monza. In other words, the places that had first been assaulted and who transferred their experience to us a week before we would have been hit”. Context 3). The creation of this system of knowledge dissemination became a support system between the different hospital contexts, working not only as a network of exchange of clinical information (“It helps you understand if what you’re doing is something that may work. If maybe you found a good solution, it may also be good for someone else who may not have thought about it. I mean, so this [set up] was there and it was a good thing, there”. Context 10), but also as psychological support (“All the ward directors were constantly in touch, so there were, let’s say, daily updates and periodic meetings simply to evaluate and see how things were going and also to discuss therapeutic approaches or how to make beds available. So from that point of view there was actually a network and there was solidarity between the various contexts. And this was important because… to discuss, to feel less alone”. Context 8). Other research [35], in fact, has shown that the comparison with other organizations allows a greater awareness and, consequently, a development of organizational learning. Professionals from different settings (6 out of 10) underlined how the construction of such a network allowed their organization to gradually improve.

## 6. Conclusions

This analysis is based on the accounts of the professionals involved first-hand in intensive care activities, exploring work practices from a dynamic social perspective [22,54,55,56]. 

The relational, communicative [57,58], and interaction dynamics between workers from the studied contexts become primary sources in understanding an organization’s development. The health emergency, in fact, besides having had a negative impact on individual psychopathologies, as highlighted by a substantial part of the literature on the subject, consented some elements to emerge within the healthcare settings that were related to organizational learning.

In this sense, it was possible to trace contextual, organizational, relational, and communicative factors that were starting conditions for the generation of organization improvement during the pandemic period. The identification of these factors by healthcare organizations can represent a valuable opportunity to reevaluate themselves, and to thus become a source of learning in governing uncertainty [59]. 

This study was based on the recognition of the nature of qualitative research and of its inherent applicability to organizations as a process of production of important knowledge, i.e., that is relevant, valid, and effective, starting from the problems that people live in real life [21,60]. Therefore, on the one hand, this article is designed as a contribution to the theories on organizational learning within healthcare contexts, paving the way to the prospect of new research on the production of knowledge subsequent to the management of an emergency by hospital organizations. Indeed, the possibility of carrying out research from the experiences of health workers offers the advantage of entering directly into the contexts under examination. Field study in this sense becomes a valuable means of understanding working and organizational processes and practices. Moreover, the value of this contribution could be valuable as the research was carried out in the period of a health emergency. Having had the readiness to initiate a study during the crisis caused by the spread of the COVID-19 virus gave insight into how professionals and organizations reacted. Specifically, it was possible to identify certain changes and transformations implemented in order to work during this difficult period. This made it possible to gather valuable data on how organizations can learn during a period of health emergency. In this way, the literature on organizational learning in health care contexts was enriched, gaining more knowledge on the subject.

On the other hand, this article represents a useful tool to give back to the hospitals that participated in the study. Indeed, it is important to make not only the professionals, but also the organizations themselves, aware of the changes that have taken place during the health crisis. This is how one can learn and treasure this difficult and disruptive work and life experience.

## Figures and Tables

**Table 1 ijerph-20-06699-t001:** Emerging themes in the Data Familiarization phase.

Theme	Definition	Example
Space Management	Possibility to create new spaces to be dedicated to COVID-19 patients and/or to modify the intensive care environments	*“Our ICU, like all ICUs, obviously expanded during COVID; a second ICU was therefore connected to ours, and strayed a bit during the first phase… After about one week… it was transferred to other operating rooms, in a different operating block and then we managed to transform a hallway, which was a waiting room for the emergency room next to our ICU, into an ICU within a week… all this was necessary in order to set up new beds. When we were at our largest we had a total of 30 intensive care beds. The change was therefore largely structural”.*
Team Reorganization	Forming working groups with new colleagues, different healthcare professionals	*“Let’s say this was a situation in which many healthcare professionals were involved at more or less different times, but many professionals cared for this patient and worked to decide what was most important at that moment, the most relevant clinical aspect to then decide how… what was most urgent at that moment”.*
Strengthened Iterpersonal Relationships	Working during the emergency period led to new, or to more consolidated, interpersonal relationships	*“But that’s how it was at that point. I saw a sacred fire, a sense of group. It was, from that point of view, very nice… the idea of having lived this experience together is as if it strengthened us all. And the disappointments were everyone’s, too”.*
Management’s Support	Helpfulness on the part of the healthcare organization and willingness to listen to the needs of the personnel	*“When the management listened to us and supported our operational strategies, everything worked great”*
Entry of Young Workers	Entry of young graduates and postgraduates in intensive care	*“So it’s inspiring to have a young person come in with this new world. Because it carries you into the new world… I am very… Let’s say, I very much appreciate it. Maybe it’s because we have very good colleagues, fantastic young people; so, their way of seeing things, their way of doing things, is extremely stimulating”.*
The Comparison With Other Realities	Exchange of information through contacts with other hospitals	*“Keeping in touch with very frequent phone calls to share as much as possible and in the first wave. We tried to do this, frantically going to all the ICUs, listening to their accounts, trying to share all the most meaningful, most important choices”.*
Process Standardization	Creating protocols and procedures to help contain the chaos	*“With COVID we standardized therapies more than ever before so as not to think: what we should and shouldn’t do. All of this should be done, period”.*
Flexibility	Learn to be flexible, adapting to constant changes	*“I think it taught us to be flexible, in every sense of the word. We really had to adapt to a lot of different things and to different aspects, to the constantly varying shiftwork and changing workplace”.*
Acquisition of Transversal Skills	Increase in skills that go beyond mere professional specialization	*“With COVID we learned to do everything, even things that did not concern our profession”*

**Table 2 ijerph-20-06699-t002:** Emerging themes in the Framework Identification phase.

Themes Extracted Deductively from the Literature	Definition	Themes Extracted Inductively from the Data Familiarization Phase	Definition
Aim	Aim shared by the team members	Management of Spaces	Possibility of creating new spaces for COVID-19 patients and/or modifying intensive care settings
Motivation	Members motivated to improve the organization’s performance	Team Reorganization	Creating work groups with new colleagues, different specialists
Safe Psychological Relationships	Feelings of trust and openness to debate among team members	Strengthened Iterpersonal Relationships	Working together in the emergency period gave rise to new or more consolidated interpersonal relationships
Infractustures	Availability of necessary resources for improving organization	Management’s Support	Collaboration by the healthcare organization and willingness to listen to the operators’ needs
Experience	Shared work experience	Entry of Young Operators	Entry of young professionals who had recently graduated or who were postgraduates specializing in intensive care
Interactions	Interactions aimed at exchanging knowledge and skills	Comparison with Other Contexts	Exchange of information through contacts with other hospitals
Shared Decisions	Sharing points of view to make decisions	Process Standardization	Creation of protocols and procedures to help contain the chaos
Intentional Learning	Organization of learning activities	Flexibility	Learning to be flexible, adapting to continuous change
Retention	Securing acquired knowledge	Acquisition of Transversal Capabilities	Increase in competencies that went beyond the mere professional specialty
Leadership	Guidance, facilitation, and support by a leader or a peer		
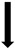	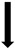
**Final Framework**
Urge felt by healthcare operators to do better-Sharing the same goal;-Motivations.
Available resources-Spaces;-Materials;-Human resources.
Shared work experienceMultidisciplinary team
Sharing points of view to make decisions-Daily moments of shared decision making.
Leadership-Management’s support;-Solid internal guide;-Guidance by more expert colleagues.
Interactions between hospitals
Acquisition of transversal skills

**Table 3 ijerph-20-06699-t003:** Example of content in the Charting phase.

Themes/Factors	Subcomponents	Context 1
Urge felt by healthcare operators to do better		The feeling of being at war pushed us to do our best.
Safe psychological relationships		Relations between ICU operators have become stronger. The other, during difficult times, was of valuable support. This has also influenced the work, making it more fluid and harmonious.
Available resources	Spaces	The organization modified the hospital’s infrastructure, creating new spaces.
	Materials	-
	Human Resources	-
Experience working together		-
Multidisciplinary teams		-
Sharing points of view to make decisions	In general	The work group’s unity ensures that decision-making processes are also shared.
	Established moments	The briefings become valuable opportunities for understanding complex cases.
Activities to disseminate knowledge		-
Standardization of practices and procedures		The creation of standardized work paths and procedures has facilitated the organization, making health professionals feel part of a well-oiled mechanism.
Leadership	Management’s Support	-
	Manager	-
	Peer Guidance	-
Relations between hospitals		-

**Table 4 ijerph-20-06699-t004:** Concluding themes in the Mapping and Interpretation phase.

Themes/Factors	Subcomponents	Context 1	Context 2	Context 3	Context 4	Context 5	Context 6	Context 7	Context 8	Context 9	Context 10
Urge felt by healthcare operators to do better		**✓**	**✓**	**✓**	**✓**	**✓**	**✓**	**✓**	**✓**	**✓**	
Safe psychological relationships		**✓**	**✓**	**✓**	**✓**		**✓**	**✓**	**✓**		**✓**
Available resources	Spaces	**✓**		**✓**		**✓**	**✓**	**✓**			**✓**
	Materials		**✓**	**✓**		**✓**	**✓**				
	Human Resources						**✓**				**✓**
Experience working together						**✓**	**✓**	**✓**	**✓**		
Multidisciplinary teams			**✓**	**✓**	**✓**		**✓**		**✓**	**✓**	**✓**
Sharing points of view to make decisions	In general	**✓**		**✓**	**✓**	**✓**	**✓**	**✓**	**✓**	**✓**	**✓**
	Established moments	**✓**	**✓**		**✓**	**✓**	**✓**	**✓**	**✓**		
Activities to disseminate knowledge			**✓**	**✓**			**✓**	**✓**			
Standardization of practices and procedures		**✓**		**✓**		**✓**					
Leadership	Management’s Support					**✓**					**✓**
	Manager			**✓**		**✓**		**✓**	**✓**	**✓**	**✓**
	Peer Guidance		**✓**			**✓**	**✓**	**✓**		**✓**	**✓**
Relations between hospitals			**✓**	**✓**		**✓**	**✓**		**✓**		**✓**

## Data Availability

Data sharing is not applicable to this article.

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
