# Peer review of "Organizational Learning in Healthcare Contexts after COVID-19: A Study of 10 Intensive Care Units in Central and Northern Italy through Framework Analysis"

_ijerph, 2023, doi:10.3390/ijerph20176699_

Round 1

Reviewer 1 Report

Dear authors, a long paper with interesting theoretical framework.

remarks:

1. make shorter sentences : see last sentence of 3.1 research objective and...

2. control references: see Lyman et al published in 2019, but in text referred to 2018.

3. without sentences out of the interviews, it is difficult to verify the conclusions

shorter sentences

Author Response

Dear reviewer, first of all thank you very much for your comments. We are pleased that you found the paper interesting. We have edited the text following your valuable suggestions. The references have been rechecked and we have re-read the paper trying to construct shorter sentences, as you suggested. Regarding point 3, in the discussion section we have included some direct quotes from the participants' narratives, collected during interviews and focus groups. In this way, from our point of view, the reader can understand the process by which we arrived at our conclusions.

Reviewer 2 Report

You could use more recent literature to explain the content analysis that has been published in the field of nursing. You are relying on just one author from 1994.

Author Response

Dear reviewer,

thank you very much for reviewing our paper and for your valuable comments. Having decided to use the framework analysis method, we have chosen to refer to the authors, Jane Ritchie and Liz Spencer, who developed it in 1994. However, what you have suggested is very appropriate and, therefore, we have decided to point out some authors who have used the framework analysis method in the health sector. In this way, from our point of view and thanks to your comment, it is possible to give the method more contextualization through more recent literature.

Reviewer 3 Report

Many thanks for the opportunity to review this inspiring and profoundly developed manuscript. The paper presents in a well-structured and methodologilly sound way the resilience of health care workers under extreme demanding conditions. At the same time the  capability of an organization to adapt to unexpected conditions is shown.  The methodology is very well suited to answer the research questions. To make the research design even more transparent, the interview guide should be included in the appendix or in the text and a table with the demographic and occupational data of the participants should be added.

Author Response

Dear reviewer, first of all we are very pleased that you liked our paper and found it interesting and well-structured. Furthermore, we are happy that you find the methodology used appropriate for answering the research questions. Indeed, we believe that framework analysis is a method of data analysis and presentation that is able to capture subjective and organizational elements in depth, exposing them in a simple and intuitive manner. Thank you very much for your valuable comments which we believe are relevant and of great use for our paper. Specifically, we felt it essential to include the interview outlines and stimuli used to facilitate the focus groups in the appendix to make the research design more transparent.  Also in the appendix, we have added a table with the demographic and occupational data of the participants, to give even more relevance to the participants and contextualize the paper further.

Reviewer 4 Report

Dear Editor

Thank you for providing the opportunity to evaluate the article entitled "Organizational Learning in Healthcare Contexts After Covid-19: A Study of 10 Intensive Care Units in Central and Northern Italy Through Framework Analysis".

I believe that this research will contribute to the literature in the field of public health.

This article is difficult by the reader because it has a different method than the research article we are used to. For this reason, shortening and simplifying the general information and method part, the more clear expression of the contribution of this study to the literature and the original value of the study will facilitate the understanding of the article.

Sincerely

Author Response

Dear reviewer,

first of all, thank you very much for reviewing the paper. We found your comments very interesting and useful in order to facilitate the reading and understand its value. For this reason, we decided to add a part in the conclusion section to further highlight the paper's contribution to the literature. Furthermore, thanks to your suggestions, we felt it was important to emphasize the original value of this contribution.

Reviewer 5 Report

Complete and pleasant to read work.

It presents two main problems in my opinion:

- The discussion is a further extension of the results section but does not confront the data obtained with other similar studies, there is no discussion in this work, it is the mere exposition of what they have achieved.

- The bibliography is referenced as APA in the text, but then uses Vancouver in the bibliography.

Author Response

Dear reviewer,

first of all thank you very much for your comments. We are extremely pleased that you found the paper comprehensive and enjoyable to read. We have edited our contribution trying to follow your valuable suggestions. Specifically, we have changed the references in the text, adapting them to the Vancouver style as used in the bibliography. In addition, we have supplemented the discussion section by inserting some references to other studies. We believe that your useful suggestions have helped our work to be enhanced.

Round 2

Reviewer 1 Report

no suggestions anymore

no comments anymore

Reviewer 4 Report

Dear editor,

I believe that the revised version of the article is suitable for publication.

Sincerely.